# Global Challenges to Public Health Care Systems during the COVID-19 Pandemic: A Review of Pandemic Measures and Problems

**DOI:** 10.3390/jpm12081295

**Published:** 2022-08-07

**Authors:** Roxana Filip, Roxana Gheorghita Puscaselu, Liliana Anchidin-Norocel, Mihai Dimian, Wesley K. Savage

**Affiliations:** 1Faculty of Medicine and Biological Sciences, Stefan cel Mare University of Suceava, 720229 Suceava, Romania; 2BK Laboratory, SuceavaCounty Emergency Hospital, 720224 Suceava, Romania; 3Integrated Center for Research, Development and Innovation in Advanced Materials, Nanotechnologies, and Distributed Systems for Fabrication and Control, Stefan cel Mare University of Suceava, 720229 Suceava, Romania; 4Department of Computers, Electronics and Automation, Stefan cel Mare University of Suceava, 720229 Suceava, Romania

**Keywords:** workflow, protection, prevention, telemedicine, medical system reorganization

## Abstract

Beginning in December 2019, the world faced a critical new public health stressor with the emergence of SARS-CoV-2. Its spread was extraordinarily rapid, and in a matter of weeks countries across the world were affected, notably in their ability to manage health care needs. While many sectors of public structures were impacted by the pandemic, it particularly highlighted shortcomings in medical care infrastructures around the world that underscored the need to reorganize medical systems, as they were vastly unprepared and ill-equipped to manage a pandemic and simultaneously provide general and specialized medical care. This paper presents modalities in approaches to the pandemic by various countries, and the triaged reorganization of medical sections not considered first-line in the pandemic that was in many cases transformed into wards for treating COVID-19 cases. As new viruses and structural variants emerge, it is important to find solutions to streamline medical care in hospitals, which includes the expansion of digital network medicine (i.e., telemedicine and mobile health apps) for patients to continue to receive appropriate care without risking exposure to contagions. Mobile health app development continues to evolve with specialized diagnostics capabilities via external attachments that can provide rapid information sharing between patients and care providers while eliminating the need for office visits. Telemedicine, still in the early stages of adoption, especially in the developing world, can ensure access to medical information and contact with care providers, with the potential to release emergency rooms from excessive cases, and offer multidisciplinary access for patients and care providers that can also be a means to avoid contact during a pandemic. As this pandemic illustrated, an overhaul to streamline health care is essential, and a move towards greater use of mobile health and telemedicine will greatly benefit public health to control the spread of new variants and future outbreaks.

## 1. Introduction

Globally, the COVID-19 (C19) pandemic exposed systemic weaknesses in infrastructures, supply chains, government preparedness and actions, human resources, and public health systems, among others. Further, the pandemic presented challenges for government health officials and administrative managers of health care systems to maintain a consistent narrative on public measures to control the spread of C19 disease. Among other setbacks in addressing the virus outbreak, it became clear that many health care facilities were ill-equipped and unprepared to manage the influx of patients and had inadequate medical and epidemiological training to properly address patient care. Overall, public health systems were not prepared to combat a novel viral pathogen that spread rapidly across the globe as containment measures were porous and inadequately implemented at the most crucial period [1]. More than two years since the SARS-CoV-2 virus emerged, it has become apparent that cooperation in information sharing among governments and health care institutions, and clear and timely communication with the public, is critical to slow the incidence of continued infection and re-emergence of the pandemic [2]. Despite this, it remains unclear whether health care measures in any one country have adapted to cope with the next outbreak.

Part of recovering from the pandemic requires restructuring public health care systems to be better prepared to manage novel disease outbreaks that overwhelmed the traditional hospital system and significantly impaired patient care quality and capacity. Thus, public health care systems need to be remodeled for efficient and capable management of emerging infectious disease outbreaks and formulated around five measures: (1) management, (2) protection, (3) containment via control and suppression of transmission, (4) information, and (5) support (Figure 1).

Understanding that there remain shortcomings in healthcare systems, initiatives are required to modernize and adapt healthcare systems. Specifically, post-C19 health care systems need to implement strategies that: (1) limit entry to heath care facilities to provide safety for patients and medical facility staff; (2) develop protocols and measures for retaining, protecting and supporting health care professionals and staff; (3) redirect non-urgent cases from hospitals to outpatient care facilities; (4) facilitate and coordinate communication among virologists, epidemiologists, point-of-care health care professionals, and health care facility staff; (5) develop best practice guidelines and legislation to coordinate cooperative worldwide action against C19 and other emerging infectious diseases.

From early on in the pandemic, protocols for treating C19 cases were constantly changing, due to the lack of information, insufficient supplies (e.g., PPE), capacities and competence (i.e., with regard to treating C19) of care facilities, availability of effective prophylactics, and social media distortion involving controversies over prophylactic measures with repurposed drugs. In some notable many cases, prior to targeted vaccine deployment began. Until vaccines were developed at a surprisingly rapid rate in late 2020, repurposed drugs led to controversial treatments such as the widespread use of the antiparasitic ivermectin in Brazil and in most of Latin America and Caribbean countries, as well as its use across much of Africa, with various reports of success in C19 treatment that did not appear to stand up to clinical scrutiny [3,4]. In other cases, drug shortages occurred due to supply chain disruption caused by the pandemic, leading to decreases in supplies of general anesthetics such as propofol and midazolam, leading to drug replacement with lorazepam and ketamine. Likewise, fentanyl availability decreased and was replaced with other analgesics, such as morphine sulfate and hydromorphone [4]. These examples indicate a problem of supply chain disruption as well as inadequate contingency plans for institutions to stock medical supplies for regular health care needs. Moreover, the pandemic limited or canceled access to alternative treatments, medical services, and medicines were less available in general, and particularly in areas communities of lower economic/class status [5]. As health care provisioning is amended to better manage this and future pandemics, need all individuals to effectively safeguard against the pathogenic spread. Evident early in the pandemic was that health care systems around the world were largely ill-equipped to manage not only patient surges in hospitals but also in the stocks and new supplies of general drugs and equipment critical to upholding regular health care system functions. As health care provisioning is amended to better manage this and future pandemics, effective planning for care measures needs to be available to effectively safeguard against the pathogenic spread.

Across Europe and in the European Union (EU) particularly, C19 mitigation strategies aimed to limit the spread of the virus to protect the public and support economies to avoid financial collapse due to lockdowns and restrictions. A primary aim for The European countries was also to support genome sequencing of the SARS-CoV-2 genome [5,6] to enable vaccine development and medicines tailored specifically to the virus. This support extended to widespread sequencing of genomic variants of the virus and the geography of its spread throughout Europe. In the EU, economic measures supported health systems care providers through financial aid distributed by the European Stability Mechanism [7]. While detailing the molecular genomics of the SARS-CoV-2 virus and the spread of new mutant forms, health care systems struggled to manage the overwhelming number of cases and were short on many critical supplies for care and personal safety.

Cases rose rapidly in countries around the world, reaching pandemic proportions on 11 March 2020 [8], although mortality and recovery rates varied, owing in part to a lack of cohesive determination of infection and C19 mortality vs. co-morbidities, as well as accurate and comprehensive testing. In terms of sheer numbers, the greatest number of infections, mortalities, and recoveries have occurred in the U.S.A., India, Brazil, France, and the U.K. (see Table 1). Although the USA and India experienced the highest numbers of C19 cases, the mortality rate has been low compared with other nations, at 1.8% and 1.4%, respectively. For example, in Romania and Bulgaria, two EU countries, C19 mortality rates have been close to twice that in the U.S.A and India. This is despite similar strategies being implemented to contain the outbreak across these different nations as well as in other parts of the world (Table 1). Overall, containment measures were leaky and likely too late to effectively stop the outbreak from occurring, in part because governments were slow to act but also because the contagion spread rapidly. Table 1 describes the measures taken by 24 countries, selected based on available published information regarding government and public health systems’ responses to the C19 pandemic.

After the pandemic was declared in early March 2020, much of the world invoked widespread adoption of face masks (particularly KN95 filter masks), which may have had one of the most significant impacts on reducing community transmission of C19, although it is difficult to ascertain the magnitude given the inherent difficulty of tracking an effect such as this. Nonetheless, the adoption of masks, whether by mandate or willingness, is correlated with reductions in burdens on specialized care units in medical units; thus, masks had the outcome of reducing excess infections [36,37,38,39], and facilitated focused care of C19 patients by lowering the burden health care systems were overwhelmed by early in the pandemic. Additional measures that have improved outcomes globally include the adoption of C19 vaccine programs [40], minimizing interactions in public environments, isolation following contact and infection, telecommuting, and minimizing/postponing travel [41]. While reducing the spread of C19, these actions also lowered the incidence rates of emergency center visits (by approximately 25%) normally seen in wards directly related to lockdown measures [42], thus reducing the burden that health care systems faced early in the pandemic and prior to the adoption of containment measures.

Since mid-2021, the effects of shutdowns on health care systems are still visible, especially in underdeveloped countries. Indeed, the past two years have been a lesson, worldwide, suggesting a vital need for medical services resuscitation and reorganization. Critical care facilities will need to develop more effective ICU triage, expandable ICU capacity and staffing pool, safer designs, efficient and sufficient supply of consumables, adequate stocks of effective PPE, devices, and pharmaceuticals, and a greater focus on the well-being of health care workers [43]. Additionally, better end-of-life care needs to be included in the management reorganization of hospital units. Lessons from this pandemic point to the importance of digital transformation in health care, as well as the reorganization and streamlining of epidemiological registries that clearly need to be part of adapting health care systems to manage the wave of this, or the next, pandemic. Reorganizing a robust health care system should be a priority for global health, and the spread of SARS-CoV-2 demonstrated this by overwhelming most systems, regardless of geography. A reorganization of health care systems can promote efficient health care by greater availability of therapeutic and otherwise life-saving drugs and personal preventative medicine, telemedicine approaches, and reduce the number of emergency ward visits and hospitalizations overall [44,45,46].

The development of vaccines and their equitable distribution worldwide alone have yet to end the C19 pandemic, yet much of the financial, research, public health, government, logistical, and human resource efforts have been involved in the design, production, acquisition, and distribution of vaccines that are not absolute in their control or spread of C19 [47]. People in countries with a vaccinated majority in the population are more likely to overcome disease symptoms with reduced hospitalization rates, although because the vaccines do not block infection [46], they do not stop transmission. As vaccines become more effective at stopping the spread of SARS-CoV-2, the creation of vaccination platforms with the identification of need-based priority groups, deployment of vaccination centers in communities and beyond, and trust in the medical professionals will be vital for effective viral control [48]. Additionally, the transparency of government authorities and agencies, research experts, and health care professionals is necessary to educate the public about the safety and side-effects of vaccines versus the risk of C19. Indeed, a system to report potential side-effects and other issues concerning immunization will help people to make informed decisions about the risk versus reward of vaccination [49,50]. While these continue to be issues two years into the pandemic, there are other needs in health care that must adapt to manage future outbreaks. Below, we discuss some of the challenges shared by health care systems around the world and discuss how the future of health care, as in other practices, has gone online, which has many advantages for both patients and care providers.

## 2. Challenges of the Pandemic for Different Specialties

### 2.1. Testing Laboratories 

The World Health Organization (WHO) reported that the SARS-CoV-2 virus can be transmitted through blood, stool, saliva, and respiration, making the availability of PPE to health care workers a critical part of treatment, containment, and health of the workers themselves. In the general public, precautions being less strict, the disease spread widely. This is where testing laboratories were vital early in the pandemic, and continue to play a crucial part in documenting the spread of C19. While testing is a post hoc measure that can be taken, it does reveal the prevalence of infection, which can provide valuable epidemiological data and help coordinate health care needs for infected individuals; rapid identification and/or diagnosis of C19 cases can lead to rapid treatment for patients, with the workflow adapting to evolving care procedures standards for infection detection [51,52].

Generally speaking, standard protocols dealing with C19 cases followed that patients were transferred to C19 containment areas, although the details of these are not investigated, meaning that it is difficult to state that there has ever been a cohesive global protocol. While public health directorates were in most cases notified of cases, it has been burdensome to trace and track contact patients had with families and other contacts, limiting population level containment approaches. While the majority of cases globally did not require hospitalizations, those patients treated at home still posed health risks because ultimately, individuals contacted others by the nature of the modern world in which the majority of people live in urban environments with close contact.

Asymptomatic people, often untested, are problematic parts of the pandemic; therefore, comprehensive testing strategies are the best solution to mitigate the spread of coronavirus. Asymptomatic transmission of SARS-CoV-2 is the Achilles heel of controlling the C19 pandemic [53,54], so continuous testing of staff that attend to populations of indigents and those in need of care is necessary. The use of rapid tests antigen SARS-CoV-2 has streamlined emergency departments and facilitated public access to home-based testing methods, although false negatives are of course possible. Despite the minor drawbacks, the benefits of rapid tests and at-home testing kits play a role in tracking the spreading impact of the virus.

According to regulations established worldwide, testing and metagenomics laboratories involved in the detection and sequencing of the SARS-CoV-2 virus need to be managed by trained staff or experts, who must comply with the circuit rules (mainly separate input and output flows) and be equipped with nucleic acids extractors, RT-PCR devices, ultra-low freezers, UV lamps for decontamination, and other disinfection equipment, automatic pipettes (robots), and contamination-free consumables [55,56].

### 2.2. Emergency Department

Unfortunately, the C19 pandemic has claimed many lives. However, it has also urged the rethought of the medical system around the world. The reorganization of the emergency department was auspicious, the models remain even after the pandemic so that the workflow is better organized, interventions are faster, and patients with minor needs or who do not require emergency treatment seek medical advice more often by telephone or telemedicine [57,58]. More than any other ward, it was the emergency department that dealt with C19. Most of the medical staff was transferred to this department, they were trained and always available. Emergency rooms were often overcrowded. The organization of the medical act on colors facilitated the decompression. For example, in Italy, a heavily affected country, the emergency department was organized by color, depending on the severity, white, green, yellow, and red [59]: red (immediate access), orange (access in 15 min), blue (access in 60 min), green (access in 120 min), and white (access in 240 min) [60]. This color sorting was later taken over by other hospitals in different parts of the world, representing a model of good practice. Emergency detection of C19 was paramount in treating the patient and eliminating the risk of transmission. Thus, in addition to continuous PCR tests, doctors have found other ways to identify the disease, by long ultrasound or just by checking the symptoms [61]. In all countries that have experienced a pandemic, the department has been supplemented with PCR equipment, CT scans, and ICU units, either by redistributing them from its own unit or by donations made by hospitals in more developed areas or for the benefit of states that have jumped to the aid of countries severely affected by the pandemic. Funders, industry, academia, government agencies, and regulatory bodies have helped emergency departments around the world, making it easier for sick people to access medical care and treatment [62]. The protective equipment was supplemented for the staff of the emergency departments (overalls, high protection masks, gloves, face shields, or goggles), and the workflow was digitized so that there were immediate connections between the reception area and the area of care and treatment. Critically ill patients were mutated in airborne infection isolation rooms or negative pressure isolation rooms, with HEPA filtration of the recirculated air [63,64].

### 2.3. Dermatology

Much of the work of dermatologists has been reorganized during the course of the C19 pandemic. Many dermatologists joined C19 treatment facilities, which left non-essential cases abandoned to focus on critical patient care. Thus, hospitalizations representing non-medical emergencies were ended and other usual consultations were conducted online through telemedicine. Where not possible, such as in melanoma cases that require surgical removal from the earliest stages, procedures were performed to protect staff from C19. Unlike other countries, where the reorganization of the workflow has facilitated the possibility of treating dermatological diseases that were urgent, in weakly developed and developing countries, many diseases that required emergency care could not be treated (solid tumors, metastatic disease, metastatic melanoma, etc.) [65].

In cases of dermatological emergencies, triage is usually essential. As a rule, dermatological consultations cannot be performed from a distance of less than 25 cm and much less in the case of dermoscopies or other interventions. In cases of patient encounters, personnel need to use PPE and be aware of contact risks. Decontamination procedures need to be strictly followed, prior to and subsequent to patient contact, particularly after direct contact with any contaminated surfaces or body fluids. The European Task Force on Atopic Dermatitis recommended continuing immune-modulating treatments, following the recommendations of the European state authorities requiring strict adherence to surface and skin hygiene protocols by replacing classic soap with non-irritating agents and using moisturizers after each application.

### 2.4. Orthopedics

Akin to most hospital wards, orthopedic wards have been completely reorganized. Interventions considered non-emergency have been rescheduled to allow major emergency and oncological cases admitted for procedures. Mild cases which would otherwise be conducted in in- and out-patient settings were postponed until they could be performed safely after the resolution of the pandemic. Among them, pregnant women, immunocompromised patients, or those over 60 years of age, were still considered medical emergencies. Medical staff used PPE to conduct intakes and procedures. Workflows were reorganized because, following anamnesis and initial consultations, it was not always possible to establish the exact status of patients, which required daily monitoring for health changes, taking into account potential for C19 symptoms such as fever, loss of taste or smell, respiratory or gastrointestinal symptoms, and cardiac irregularities [66]. Suspect nasopharyngeal exudates were immediately collected from patients for PCR testing and moved to designated areas, where they remained until results were received [67]. Patients positive for C19 who represented medical emergencies were transferred to designated containment areas where surgical procedures were performed. The number of physicians entering operating rooms was limited, and aerosol-generating procedures were avoided [68]. Where C19 patients received treatment equipment (monitors, computers, ultrasound, etc.) were required to be shielded from contamination and facilitate cleaning to reduce contamination risk [69]. Postoperative routines were limited to maximum capacities, and when possible, portable radiography equipment was used and disinfected immediately after use. For post-operative care, easily changed dressings or splints were mostly used.

### 2.5. Obstetrics and Gynecology

SARS-CoV-2 is problematic for more than 100 million pregnant women worldwide [70,71]. Due to suppressed immunity, they can develop moderate to severe forms of infection that can also affect the fetus. Pregnant women with C19 have an increased risk of miscarriage, premature birth, and preeclampsia. Fetuses are at higher risk of mortality (2.4%), neonatal mortality (2.4%), or requiring intensive care [72]. As with other patients, C19 screening is very important. Pregnant women who contracted C19 but did not show respiratory symptoms or other symptoms were quarantined at home, remaining in contact with their primary care providers. 

The health of pregnant women must always be kept under supervision, necessitating continuous testing. Blood tests that follow various important parameters should not be neglected. In the most affected areas, it has been suggested that intakes be started at home, and then women come to the hospital to avoid unnecessary exposure to patients. Usually, intakes underwent hospitalization for 1–2 days prior to birthing, but during the pandemic, their contact with medical units was limited to provide safety for the mother and fetus. These situations can present complications with timing, as it creates scheduling problems for hospitals and patients.

During cesarean section, anesthesia can be performed with an epidural, but it is recommended to limit the use of nitrous oxide due to the risk of generation aerosols, which risk the spread of SARS-CoV-2. Breastfeeding is recommended for women infected with C19 because according to the latest studies, both IgG and IgM antibodies are transmitted through breast milk [73,74,75]. Antibodies are present in breast milk as early as 2 weeks immediately after vaccination of mothers, being transmitted to breastfed infants [73,76]. Visits have been banned because access by outsiders is a common way of contamination, using the internet for online dating instead [77]. Before discharge, both mothers and newborns are tested and can leave the hospital only after testing negative for SARS-CoV-2.

### 2.6. Pediatrics

Unfortunately, the pandemic has also hit the pediatric sector. If at first, the number of cases of children was not so high, now, due to the new variants of SARS-CoV-2, children have become the target of the disease. Fortunately, children’s symptoms are less severe, such as fever, dry cough, nasal congestion, abdominal discomfort, or diarrhea [78], often asymptomatic. Unfortunately, there have been cases that required pediatric emergencies. Thus, there were beds in the pediatric infectious area and all hospitals were prepared for pediatric emergencies. Most children in need of emergency medical care experienced moderate to severe respiratory infections: influenza and bronchiolitis, meningitis, sepsis, osteomyelitis [79], and asthma [80]. Omicron, the new SARS-CoV-2, is much more contagious and prevalent among children, although the effects of the disease are greatly diminished, hospitalization and ICU utilization and mechanical ventilation for patients are less than in the case of infections with Delta variant predominated [81].

Telemedicine has been a lifeline for pediatrics. The doctors kept in touch with the parents of the children who did not need medical emergencies, and they received all the indications without going to the hospital, where the risk of infection was higher [82,83,84,85]. Where it was necessary to present to the emergency room, additional measures were observed, both for the medical staff and for the patient or his relatives: masks, infrared thermal screening, special spaces for emergencies, and “clean area”, restricting patients in operating rooms and banning them in the common play area [86,87,88], or their differentiation according to the color of the bracelets received at the entrance (for example, yellow for suspected C19 route and white for standard route) [89].

Often, young children are accompanied by parents or legal guardians, so the risk was higher. Therefore, in order to eliminate any risk, the children were allowed to be accompanied only by a caretaker who was present during the procedure, important for the physical or emotional well-being and care of the pediatric patient [90]. Unfortunately, many children no longer have access to advanced health care or scheduled procedures in more developed countries than their home state due to travel restrictions and the reorganization of hospitals, and the transformation into C19 units [91].

As the generated pandemic does not seem to be ending, it is mandatory to institute a series of measures, not only by reorganizing the flow in pediatric units, but also by reorganizing the protection measures in schools and the way of attending courses, a renewed concept of the health system, or telemedicine [92].

## 3. Medical Workflow Reorganization

The pandemic forced health care systems to not only reorganize hospital wards and units, and medical and auxiliary staff but to also transfer equipment and supplies from other departments for use in treating patients with C19 [93]. Respiratory equipment (e.g., ventilators, pumps and monitors, materials, PPE, and medications were transferred to sections in greatest need [3]. However, anemic infrastructures and economic conditions in many countries limited access to and availability of protective equipment and PPE supplies, rapid test kits, and availability of necessary medical care. Global disparities in economic conditions left health care workers in low and middle-developed countries at greater risk of exposure to the virus than those in developed countries who had greater access to advanced treatments, materials, and technologies [94]. Even within countries, regional disparities in economic conditions created inequalities in treatment, and therefore the outcome of C19 cases.

In operating rooms, when interventions could not be postponed, workflows were reorganized to prevent contamination at each step leading to operating rooms. For procedures on patients diagnosed with C19, entrance to operating rooms was restricted to anterooms with lower atmospheric pressure with airdrop seals [95]. Other measures aimed to reduce auxiliary medical staff involved in procedures (e.g., maximum of two nurses and an anesthesiologist) and uses additional PPE and powered air-purifying respirators [95]. These precautionary measures allowed the flow of interventions/procedures to continue under the looming threat of possible transmission of C19.

A very important step is to sort the patients from the moment they arrive at the medical unit (Figure 2).

Most health care units conducted the workflow in separate areas, usually located near hospitals, and only patients who presented negative C19 tests were admitted [96] and remained in isolated areas for up to 72 h (Figure 2).

Although patient temperatures were checked upon arrival at hospitals and they completed epidemiological questionnaires, PCR testing was required. Testing temperatures alone are not conclusive for coronavirus infection; other medical factors and conditions are quite commonly associated with raised body temperatures, such as in cancer patients where fever is a common therapeutic consequence. Conversely, asymptomatic C19 cases are always a risk as well as pre-symptomatic individuals [97]. Positive coronavirus cases were redirected to specialized treatment areas [98,99].

## 4. Medical Staff Protection

Protective equipment includes gowns, medical protective masks, goggles, gloves, disposable cap masks, disposable clothing, and full-face holds [100,101,102]. Washing hands with alcohol have become a basic safety measure [103,104]. The medical staff was trained to wash their hands properly and to use the protective equipment correctly (for example, avoiding shaking the equipment at the time of disposal) [105]. According to tests, the virus responds better to ethanol than isopropanol, at a contact of at least 30 s. Frequent use of antiseptic chemicals, and protective equipment for a long time has led to health problems among medical staff, such as irritations, facial inflammatory papules, urticaria, rosacea, seborrheic dermatitis [18]. Where there were already skin lesions, they were aggravated. A solution in this regard was the replacement of towels with paper napkins. Even the use of personal mobile phones has been restricted in units treating C19 patients or, where this has not been possible, cell phones have been used. The use of shelter hospitals for infected patients has reduced the spread by minimizing contact with healthy people who come to the medical unit with other health problems [106]. All these measures were aimed at reducing the risk of contamination between the affected and outer areas [107].

Some states have developed support programs for medical staff to regularly check on their health and provide psychological support, and provided online platforms for tracking health and communications [108]. Medical students have been trained in the fight against the pandemic, being distributed in the staff of public health units, facilitating the connection between patients and the medical unit, and providing moral support and assistance for health workers and the population [109].

In operating rooms, since procedures such as laparoscopic interventions, endoscopes, or intubations generate aerosols and can easily transmit the SARS-CoV-2 virus, for the safety of medical staff it was decided to postpone those who do not represent an emergency [110]. During the pandemic, the rotation of personnel in areas with maximum risk was applied, in order to minimize exposure and eliminate possible contamination due to frequent contact with sick patients. This also helped to eliminate burnout and mental breakdown.

### 4.1. Psychological Effects of the Pandemic in the Health Care Environment

The massive numbers of ICU cases presented during the pandemic that overwhelmed global health care systems led to psychological burden and fatigue of medical care workers [111]. Beyond health care workplace conditions, lockdowns and quarantines, shop closures, limited access to food and resources, school closures, physical isolation, restricted movement, altered daily routines, and perhaps most importantly, the loss of social and family life have exacerbated psychological and physical burdens on health care workers [112]. As depression, fear and anxiety emerged among many people during the pandemic, support for medical staff and patients presented new challenges. Isolation in C19 wards led to a new need for patients during hospitalizations, namely psychological support, which required health care workers to address patient psychological needs directly related to conditions that arose due to the pandemic. As this issue emerged, participation in mental health training [113], access to therapists, psychologists, and psychiatrists, support and educational materials, and organizing support teams all became necessary to the continued function of health care systems [114]. The goal was to ensure a climate of communication and collaboration among health care workers, where teams were set up to collect screening data to adapt the workflow to the continuously changing situation presented by SARS-CoV-2, and to provide workplace support for fatigued staff [115]. The pandemic negatively affected morale in health care environments, with cases of post-traumatic stress disorder and acute stress reactions that affected all staff ranks, including administrative as well as support staff [116,117,118,119]. What showed to be critical to stabilizing morale were collegial and familial support systems, religious institutions, and proactive management teams [115,120]. In addition, health care workers were endowed with financial compensation to help ameliorate part of the psychological burden and help workplace morale [121].

Following the guidance in recommendations provided by the WHO, government, and health care institutions, managing pandemic stress in the health care environment requires focusing on one’s own needs and those of others, limiting exposure to social media and the news, preventing the dissemination of misinformation, resting and keeping health sleep habits, and balanced nutrition, exercise, and relaxation or meditation.

The role of the media in transmitting information proved quite destructive to relaying fact-based messages during the pandemic; many television programs promoted negative and false news that induced confusion over facts and panicked the population at the start of the pandemic. Medical workers routinely struggled to overcome misinformation presented in media channels, which exacerbated caregiver fatigue. Various self-trained “specialists” provided erroneous information on how the virus spreads, treatment methods to deal with the pandemic (e.g., consumption of hot water, snake oil, or even silver), and even anti-vaccination information [122]. Thus, in many countries, the vaccination campaigns and containment measures have failed to stall the virus for nearly two years since it emerged. Low vaccination rates may be in large part due to social media misinformation, and ineffective presentation of facts by government and health care leaders. While developed countries have relied on accredited experts to promote the benefits of vaccines, in developing countries social media has played a largely negative role in promoting misinformation. Ineffective communication of facts and misrepresentation of information caused panic, instability, and mistrust in health care systems [123]. Thus, the rapid and far-reaching spread of accurate and inaccurate information, “infodemia”, created a new challenge to control the C19 pandemic [124].

### 4.2. Telemedicine

The agglomeration and transformation of medical units into C19 areas prevented access for many in need of medical services. The situation called for urgent action, and telemedicine emerged as an important tool for patient care. In 2010, the WHO established the Global Observatory for eHealth (GOe) which oversees the benefits of applying information and communication technologies in medicine [125]. According to the WHO, telemedicine, or remote treatment among many other terms, facilitates video communication between patients and care providers to share information on diagnoses, treatment, and prevention of disease, and for evaluation and research [126]. Telemedicine has many advantages because it can save time and resources while providing clinical support, and it increases access for patients where travel to a care facility is prohibitive [127]. In addition, patient contact with care providers can strengthen confidence in the medical system and avoid overcrowding of medical units for problems or conditions that do not require in-person visits. Telemedicine, much like work from home, means that routine checks can avoid exposure to coronavirus in hospital and outpatient settings.

Mobile applications in telemedicine allow file-sharing of vital statistic measurement and monitoring [128,129].

First of all, access to medical services can be made from anywhere, regardless of whether the patient or the doctor is at home or away. Moreover, the time lost in the waiting rooms is no longer a problem. The method could also help patients be more honest than if the meeting had been face-to-face. The problems that may occur refer to the lack or instability of the internet connection, the lack of training in using the networks, or the impossibility of accessing a computer or other device necessary for the online meeting. Another advantage of using medicine refers to cost savings, both for the patient and for the medical staff or units [130]. Eliminating geographical boundaries is perhaps the most important benefit of telemedicine. In this way, access to medical services abroad is no longer a problem, and consultations can be made for people who are unable to move or travel, cannot reach the hospital, or are in a situation that requires immediate consultation.

During the C19 pandemic, telemedicine played a very important beneficial role, as it eliminated the risk of infecting the population requiring treatment or regular medical check-ups, facilitated access to information and identification of the health status of the quarantined population, facilitated multidisciplinary interventions, with medical teams from various parts of the world, in order to exchange experience and establish optimal solutions [131,132]. Therefore, many areas around the world, such as China, USA, and Australia, have established national telemedicine centers and regulated a series of measures to establish a legislative framework [133,134,135,136,137,138].

In recent years, the adoption of mobile apps and online platforms in health care has expanded patient access to medical providers. Various health care interface applications are becoming widely available in many parts of the world, connecting patients with medical care facilities that deliver real-time results for diagnostic tests, health care minutes with care providers, and other health measures. This became more important as the C19 pandemic disrupted the longstanding tradition of patient care, as patients no longer had access to medical services as before, and therefore digital solutions increased in importance for basic health care problems. The development of digital health solutions has in many ways revolutionized health care by streamlining patient consultations with medical professionals via telemedicine. For most basic care needs, telemedicine removes contact risks in care facilities, which is particularly important for the immunocompromised, as well as reducing the spread of contagions, while it also eliminates waiting rooms, and reduces travel costs, travel time, and lost work time, among many other benefits. Telemedicine is indeed a valuable platform, but it cannot of course replace health care that requires patient visits for clinical observations, specialized interventions, and exams [139].

With digital applications, patients can readily access personal health information and diagnostic results, have real-time consultations with primary care providers and specialists from far away distances, have access to prescriptions and medication dosing information, receive care alerts initiated by providers, and participate in various support groups important for recovery. As mobile device usage (i.e., smart phones, smart watches, and tablets) now outpaces computer use around the globe, the development and adoption of mobile health care apps are increasing access to health information. Mobile health apps are becoming more specialized for specific diagnostic tests and can measure various parameters with the attachment of external devices, such as inhalers, digital stethoscopes, and blood pressure monitoring [140,141]. The benefits of digital health have resulted in as many as 318,000 medical health apps developed as of 2017 (IQVIA Institute for Human Data Science, The Growing Value of Digital Health: Evidence and Impact on Human Health and the Healthcare System [142], which has brought the mobile health (“mHealth”) market upwards of 8.0 billion dollars in 2018 and is expected to exceed 111 billion dollars in 2025 [143,144].

These technologies are also supported by the U.S. Food and Drug Administration, which, since 2017, has issued the Digital Health Innovation Action Plan [145]. Its purpose is to encourage digital health innovation that includes three aims: (1) new guidance regarding the regulation of digital health, (2) developing new regulatory approaches to oversee digital health, and (3) building expertise on digital health within the agency.

Worldwide, for the safety of medical care, mHealth applications should be monitored and regulated by oversight boards of governmental health agencies. For example, in the United States, laws and regulations from three federal agencies regulate mobile health applications:1The Office for Civil Rights (OCR) within the U.S. Department of Health and Human Services (HHS) enforces the Health Insurance Portability and Accountability Act (HIPAA) rules–to protect the privacy and security of health information;2FDA enforces the Federal Food, Drug, and Cosmetic Act (FD&C Act)–regulates the safety of using medical devices, and eliminates the risk when health app does not work properly;3Federal Trade Commission (FTC) enforces the Federal Trade Commission Act (FTC Act)–prohibits or creates alerts of unfair acts or practices, and monitors apps’ safety or performance.

During the C19 pandemic, many governments around the world-initiated strategies for the development of trace contact mobile alert apps. For example, the National Health Service in the United Kingdom developed an app to warn when physical distancing violated the C19 health care guideline, or when close contact with a positively diagnosed person occurred. Within ten days of development, the app was downloaded over 10 million times. Another measure initiated by the World Health Organization is a designated health alert messaging service accessible via WhatsApp that provides the latest news and information on C19 that includes details on symptoms and how people can take precautions for C19 and other public health outbreaks. The health alert messaging service also provides up-to-date situation reports helping government decision-makers and health care agencies take effective measures to protect public health. To date, as many as 300 apps have been developed that focus on curtailing the spread of C19, although the effectiveness of these apps varies widely, requiring further evaluation [146]. Certainly, mHealth apps that focus on C19 will be optimized and others developed, as the utility proved extremely useful during the pandemic (Table 2).

## 5. Conclusions

The SARS-CoV-2 pandemic illustrated demonstrably that microbes pose a grave threat to global population health and financial systems. Unfortunately, although there are means available to mitigate C19 effects, many around the world are reluctant to take the vaccines available because of the infodemia around the SARS-CoV-2 virus. Countries, where the majority of the population are vaccinated, have begun to recover economically, but many countries still lack specialized drugs to effectively treat people and begin an economic recovery. Globally, the reorganization of legislative and medical measures has the effect of limiting the spread of SARS-CoV-2, reviving economies, and reviving activities that were slowed or stopped as containment and protection measures. As each country has had its own course of action, the relaunching of strategies for recovery varies and can inform other approaches. New and improved ways of approaching medicine, namely telemedicine, have advantages that will likely make it a default platform for the foreseeable future. New measures and innovative approaches will certainly continue to emerge as SARS-CoV-2 will be present in the global population hereon. We highlight the importance of mobile health apps and telemedicine in the reorganization of medical systems to improve public health as the world community will likely face new pandemics and outbreaks similar to C19.

## Figures and Tables

**Figure 1 jpm-12-01295-f001:**
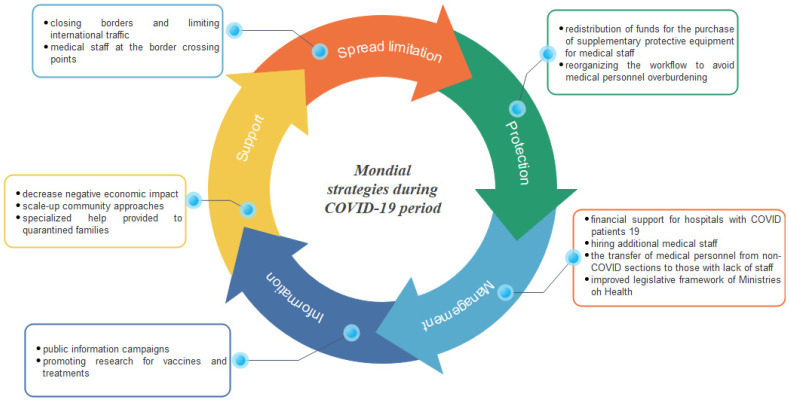
Mondial strategies during C19, describing five measures that focus on redesigning public health care systems to better manage future pandemic events.

**Figure 2 jpm-12-01295-f002:**
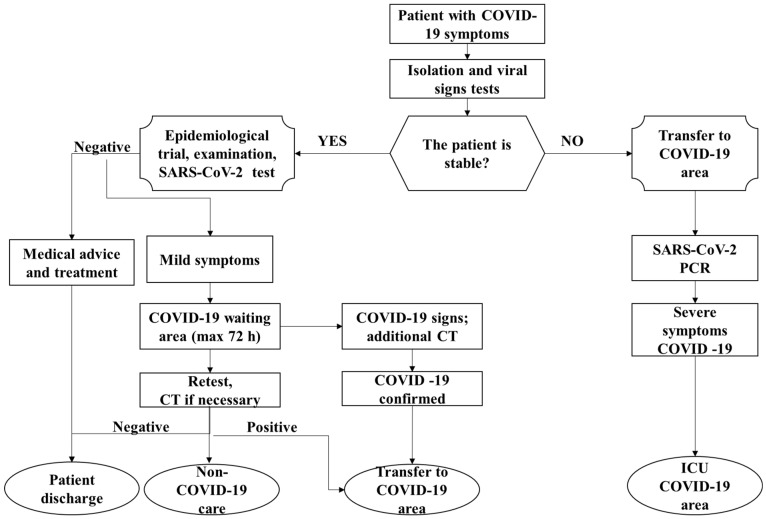
Pandemic workflow for hospital intake procedures.

**Table 1 jpm-12-01295-t001:** A select list of nations that highlight approaches that nations took to manage C19.

Country	Measures	References
Australia	Australia’s early physical distancing measures, stable political system, stable wealth, and geographic isolation may have contributed to its relative success in managing the C19 pandemic. Widespread public support for physical distancing measures and the government’s financial support for individuals and businesses afflicted by the pandemic meant that these measures could be quickly put in place. On the other hand, confused and inconsistent communication, especially in the early stages of the pandemic, detracted from government efforts to manage its response. The pandemic exacerbated existing social inequalities, highlighted by racism and dependence on import industries. Measures to reduce social inequality through secure employment are likely to be juxtaposed against measures to contain the costs of employment in forthcoming policy debates. The impact of the withdrawal of job placement schemes while unemployment remains high will likely exacerbate social inequality. The Australian health system’s response to C19 attempted to manage the spread and increase due to its limited number of intensive care unit beds across the continent nation, and the limited supply of ventilators, masks, and personal protective equipment (PPE). New skills and ways of working were required within the health system: these included contact tracing, telehealth, and leveraging resources from the private hospital and health sector. The Australian Therapeutic Goods Administration instituted procedures for rapid deployment of a range of medical devices used in the treatment of the disease, including ventilators and point-of-care testing kits, and increasing the availability of PPE.	[9,10,11]
Belarus	Government authorities initially denied the virus was a public health hazard. Later, they announced that drinking vodka and working in the fields offered protection from C19. Due to its official rejection of the pandemic, Belarus did not impose any quarantine measures, did not restrict cultural activities, events, or the retail industry, and instead operated on a campaign to ignore the existence of the SARS-CoV-2 virus. Although geographically isolated and distant from pandemic epicenters in western Europe, C19 finally reached Belarus, forcing measures for self-quarantine, social distancing, mask-wearing, and avoiding shops and public gatherings.	[10]
Brazil	The C19 pandemic created hardships for developing countries such as Brazil. From January to March 2020, the pandemic reached crisis proportions that exacerbated political, social, and economic problems. However, Brazil also reaffirmed its leadership and coordination capacity, especially in fiscal and economic measures, while the number of healthcare jobs decreased. In large part, healthcare workers were supplied with the necessary PPE following WHO recommendations. However, most healthcare workers did not receive proper training for treating patients suspected of coronavirus infection. Physicians and nurses were overworked and suffered fatigue. Many healthcare workers reported difficulty sleeping as a result of pandemic stress and workplace fatigue.	[12,13]
California	As a case example of a state in the U.S. where it is difficult to oversee a federal role, following the establishment of Critical Care Services and public health guidelines, C19 patients received better care, and mortality dropped to 0.008%. The public health guidelines involved social distancing, hygiene education, widespread C19 testing, acquiring a substantial inventory of ventilators, PPE for healthcare workers, and ample therapeutic drugs and pharmaceutical supplies. The expansion of resources, including ICU capacity, trained staff (mainly physicians, physician assistants, nurses, and respiratory therapists), and supplies expanded from 20% to 100% in the contingency and upwards of 200% during the peaks of the C19 crisis.	[3]
China	China responded rapidly and effectively to contain the virus within the Wuhan province where the outbreak began and quickly recovered due to social contact restrictions that were strictly observed. The success was achieved by rapid establishment of lockdowns and construction of modular hospitals, use of state-of-the-art equipment for population-level diagnostics, recruitment of the best health workers, systematic population screening with testing and isolation, prevention of nosocomial transmission, the development of two vaccines, and subsequent administration of an unparalleled vaccine campaign.	[14,15]
Finland	Finland was the only country to initiate a hybrid strategy to control the C19 spread by shifting from large-scale restrictive measures to more targeted pandemic management measures. Border entry restrictions excluded outside visitors, while Finnish citizens were required to remain in a 2-week quarantine upon re-entry. Non-essential retail operations were closed, while essential retail was allowed (e.g., grocers and pharmacies). Educational institutions suspended on-site activities. The public health institutions and small businesses were financially supported to relieve economic consequences; consulting services that support health care and business were developed.	[10]
France	Following the well-publicized case explosions in Italy and Spain, and despite it being a foreseeable event, France failed to advance stocks of medical supplies, PPE, and tests for SARS-CoV-2 detection. As cases mounted, the government enacted measures to restrict public gatherings, and the operations of restaurants, shops, schools, and non-essential activities; however, the government funded essential activities, focusing on supporting the healthcare system.	[10]
Hungary	In March 2020, authorities declared a state of emergency and adopted a law to enact restrictions without the oversight of the parliament. Government communication with citizens was inadequate, and therefore heavily criticized by the population, leading to non-compliance by the citizenry. Financial measures were created to support businesses and entrepreneurships by tax abatements and accelerated VAT refunds as a means to ameliorate the economic impact of restrictions.	[10,16,17]
Iceland	The primary healthcare in Iceland managed to accomplish its role as a first-line gatekeeper and was able to change its strategy swiftly in an effort to deal with C19. At the same time, traditional maternity and well-child care was preserved. The use of primary healthcare for non-C19-related issues decreased, indicating substantial flexibility in the organization. Iceland has been lauded for its approach to handling the virus, which has led the way in terms of the gathering of scientific evidence and its implementation in policies. Iceland has used the resources of deCODE, a private sector genetics firm located in Reykjavík, in tandem with the public health services to track the health of every individual in Iceland who has tested positive for the virus and, uniquely to this nation, sequenced the genetic material of each viral isolate and screened more than half of the nation’s population for infection. This information has informed the recommendations of the chief epidemiologist in Iceland concerning border controls and domestic restrictions.	[18,19]
India	An ill-equipped infrastructure and anemic pool of public healthcare professionals led to major failures to slow the C19 spread. Lockdown periods were extended, and measures were taken to equip care centers with C19 facilities, increase the ranks of trained healthcare professionals, and provide the population with PPE.	[20,21]
Italy	Initially one of the most affected countries in the world in early 2020, Italy altered strategies to focus on reorganizing medical departments and supplementing intensive care beds, closing/blocking activities considered non-essential, financially supporting businesses, and creating isolation areas that reduced risks to healthcare staff.	[22]
Japan	The Japanese understood the importance of self-quarantine, the telemedicine services worked intensely, being useful to people who could be treated at home, the medical staff understood and respected the rules of protection.	[23]
New Zealand	New Zealand adopted a set of non-pharmaceutical interventions aiming to bring C19 incidence to zero. The transmission chains were spread out across the country, with the highest incidence in popular tourist areas, and large transmission events such as weddings led to transmission chains containing multiple age groups. The reconstruction of detailed epidemiological links is paramount to improving understanding of the spread of SARS-CoV-2 and keeping close surveillance on settings with a high risk of transmission.	[24]
Pakistan	The containment measures included self-isolation, social distancing, restricting public gatherings, supplementing public health facilities and staff, concentrating human resources on areas treating patients with C19, providing necessary resources and equipment, and mental and economic support for the population.	[25]
Romania	A disorganized and under-funded healthcare infrastructure coupled with poor organization by government authorities caused serious failures to address public health and led to a widespread and rapid outbreak. A lack of medical professionals trained in infectious diseases created a massive shortfall in C19 patient care. Once a State-of-Emergency was declared, non-specialist physicians and healthcare workers were deployed to staff C19 sectors. As a result, healthcare workers came in regular contact with a large number of C19 patients, which magnified contamination risks and further strained the healthcare infrastructure. Health-care workers were not provided sufficient Personal Protective Equipment (PPE) and worked overtime without adequate rest, which led to declines in performance from illness, stress, and fatigue, which essentially broke the healthcare system. As in other nations, non-C19 patient care needs were reclassified and thereby canceled or postponed, further complicating the public health situation.	[26,27,28]
Russia	Russia initially denied the severity of SARS-CoV-2. The government sent aid to other areas affected earlier in the pandemic (e.g., Italy and Serbia). Subsequently, as Russia faced its own C19 outbreak, they ran out of necessary supplies to combat contagion spread. Soon thereafter, Russia closed its borders, first to China, and then to all foreigners, but throughout it appeared more concerned with financial stability than the public health crisis.	[29]
Spain	In the beginning, testing was not provided to health workers that had been exposed to patients with C19, resulting not only in dangerous conditions for workers themselves and the people in close contact with them but also for other patients hospitalized for conditions not related to the C19. Another consequence of the lack of testing was the possible underreporting of cases, resulting in overestimated mortality rates being reported due to the lack of certainty on the real number of positive cases of C19 in Spain. Finally, the lack of surveillance and case detection potentially caused the further spread of the disease, as many of the unidentified cases did not follow recommended isolation measures. Overall, the lack of testing resulted in a symptom-based strategy to control the disease, which was unlikely to succeed at stopping disease transmission due to the characteristics of SARS-CoV2, which had been reported to cause a high proportion of asymptomatic and mild cases. In addition to the consequences seen in the healthcare sector, government interventions also had severe impacts on the public. These impacts included the economic recession generated by confinement measures that caused an unprecedented situation which is predicted to have multiple short and long-term effects on the Spanish economy. Additionally, psychological consequences occurred due to the restricted freedom, decreased social contact, and persistent insecurities caused by the health threat and the control measures; children, adolescents, and young adults are particularly vulnerable to these consequences given the important role of socialization. The outbreak has had a severe social impact; senior citizens, children, and women at risk of violence, families, and individuals at risk of poverty, migrants, socially excluded groups, and people with low-paying or informal jobs are some of the groups that have been severely affected by the psychological, economic, social and health consequences of the pandemic and the measures to control the spread of the outbreak, aggravating the existing inequalities across the population. To be able to meet the needs of the epidemic, health professionals decreased their regular activities at the hospital to focus all their working capacity on tackling the C19 crisis, decreasing the capacity of non-urgent and specialized medical services.	[30]
Sweden	Based on advice from the national epidemiologist, the government elected few social restrictions except for border controls. Instead, it opted for a herd immunity approach to achieve seroconversion. It is unclear if this approach succeeded, although the mortality rate remains below 1%, while still higher than other Scandinavian nations with Norway the lowest by a factor of 10.	[31]
Switzerland	Medical resources were strengthened by public-private partnerships that increased isolated treatment areas for C19, separate from other patients; supplementation efforts were extended to aid medical staff and increase availability of medicines that enhanced patient care.	[22]
Taiwan	The containment of SARS-CoV-2 dissemination was very effective and the population infection rate stopped at the level of hardly more than 0.6 percent. Strict border control and the effectiveness of contamination programs have isolated the pandemic in a few local niduses, mainly in Taipei and Taoyuan.	[21]
United Kingdom	As the number of cases rose rapidly, government measures focused on patient care and providing medicine, such as PPE, ventilators, and intensive care units. Although a lockdown was enacted in March 2020 and was viewed with some skepticism, the measure came late as cases continued to rise and overwhelm the healthcare system. Researchers and physicians collaborated in drug discovery to develop the Oxford–AstraZeneca C19 vaccine.	[32]
United States	Although access to information is high, the government lost touch with the population during the pandemic, and disinformation caused mistrust in public health authorities and advocates of protective measures such as vaccination campaigns. Much of the population grew fatigued over mandates and contradictory statements regarding C19, polarizing the people into two fractions, one of which ignored mandates of quarantine or public distancing. The most common systemic problems worsened during the pandemic, as inadequate supplies and an ineffective distribution system led to increased community spread. Perhaps the greatest failure of the U.S. at the onset of the pandemic was the delayed restrictions on international travel, which allowed individuals afflicted with C19 entry into international airports, notably in Seattle and New York.	[33,34]
Ukraine	The healthcare system was unprepared and therefore overwhelmed. Authorities were forced to adopt a pandemic response and instituted a quarantine on 12 March 2020, eight days after the first case was documented. The government appealed to the private sector entrepreneurs and businesses for monetary assistance in managing the economic and public health crisis. This internal aid provided the necessary equipment and PPE and supported healthcare facilities. After restrictions were lifted, widespread adoption of masks was put in place, public events were restricted, and the over 60 populations were advised to isolate themselves from public exposure. The government facilitated and supported small businesses, cut interest rates, and substantially increased the health sector budget.	[10]
Vietnam	A number of countries in East and Southeast Asia managed C19 spread quite effectively, notably by promoting hygienic practices to prevent spread, specifically personal hygiene, and food and water hygiene.	[35]

Included here are island nations, which due to their geography, were better able to implement containment protocols. Sweden is also included as a notable nation that rejected social control measures taken by the majority of nations in an attempt to achieve herd immunity.

**Table 2 jpm-12-01295-t002:** Examples of mHealth apps developed by government agencies for contact tracing and information delivery to curtail the spread of C19.

Country/Region	App	Type of App	Impact	References
Australia	COVIDSafe	tracing	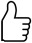 epidemiological surveillance, date, time and duration of contact 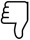 user data privacy concerns, lack of trust, ethical issues, security vulnerabilities, technical constraints	[147,148]
France	StopCovid
TousAntiCovid
Germany	Corona-Warn-App
Globally	TraceTogether
COVID Trace
NOVIDShare TraceSafe2	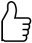 the user receives a unique ID, and at the time of infection, he is redirected to medical applications	[149]
Italy	Immuni	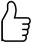 epidemiological surveillance, date, time, and duration of contact 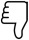 user data privacy concerns, lack of trust, ethical issues, security vulnerabilities, technical constraints	[147,148]
Spain	AsistenciaCOVID-19
Switzerland	SwissCovid
Globally	HowWeFeel	medical	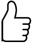 for health care professionals and researchers; a useful tool for patients, quarantine advice, and measures to ensure well-being. The authorized access was ensured by firewalls, antiviruses, and cryptographic algorithms	[149]
Mexico	Sofia	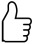 video consultations: internal medicine and pediatric consultations, prescriptions, follow-up indications; high levels of patient satisfaction, versatile and convenient tool to manage the situation	[150]
USA	Teladoc	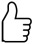 access to low-cost and high-quality doctors, clinical expertise, virtual care for consumers and clinicians	[151]

The limitations refer to mobile device ownership or accessibility to mobile apps, network connection problems, misdiagnosis of C19, or app takeover by the pharmaceutical industry [152].

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
