# Peer review of "Global Challenges to Public Health Care Systems during the COVID-19 Pandemic: A Review of Pandemic Measures and Problems"

_jpm, 2022, doi:10.3390/jpm12081295_

Round 1

Reviewer 1 Report

The manuscript has been improved on some points.  Now it is clear the overall motivation, the presentation, and review of the significant challenges that C19 presented to healthcare systems in different nations, and how they deal with them.

The review is interesting, but more work on how to apply these results to the field the special issue covers, challenges and opportunities for health information technologies, is needed. A re-orientation of the results and conclusions sections might be necessary to improve and aling the contribution to the field.

Author Response

Esteemed Reviewer:

We greatly appreciate the opportunity to have our work reviewed by you. The following phrases include the referee comments and our responses.

The manuscript has been improved on some points.  Now it is clear the overall motivation, the presentation, and review of the significant challenges that C19 presented to healthcare systems in different nations, and how they deal with them.

Response: Thank you for your review.

The review is interesting, but more work on how to apply these results to the field the special issue covers, challenges and opportunities for health information technologies, is needed.

Response: We added content specific to mobile health and telemedicine to address this comment (lines 599-673 – track changes– all markup or lines 502-572 – track changes – no markup).

A re-orientation of the results and conclusions sections might be necessary to improve and align the contribution to the field.

Response: We previously reorganized some of these sections. This reorganization is difficult to follow from the version you reviewed because of track-changes from the previous review.

Reviewer 2 Report

Firstly, thank you the authors for revised manuscript. I think my suggestion will made the manuscripts complete detail of the study and make reader to understand better. After reviewed the revised manuscript, I think this paper ready for publication.

Author Response

Esteemed Reviewer:

We greatly appreciate the opportunity to have our work reviewed by you. The following phrases include the referee comments and our responses.

Firstly, thank you the authors for revised manuscript. I think my suggestion will made the manuscripts complete detail of the study and make reader to understand better. After reviewed the revised manuscript, I think this paper ready for publication.

Response: Thank you. We appreciate your effort and time.

Reviewer 3 Report

The manuscript is marked up rendering it useless to review. Authors need to submit a finalized manuscript without markups.

The abstract is too verbose and seems like an introduction.

Mobile apps; blockchain etc., can be examined.

Table 1 - too verbose. Organize by keywords or standardized phrases. Also, consider organizing by region, developed v. developing countries, etc., rather a random collection of countries.

Consider discussion political, WHO, limited resources, haves and have nots etc. that have impact on policy.

Author Response

Esteemed Reviewer:

We greatly appreciate the opportunity to have our work reviewed by you. The following phrases include the referee comments and our responses.

The manuscript is marked up rendering it useless to review. Authors need to submit a finalized manuscript without markups.

Response: We submitted this manuscript with track-changes as requested by the editorial office. The version you were given included changes to a previous submission. You can view the manuscript without track changes markups. We had organized Table 1 by country/region.  We apologize it was difficult for the review process, but it was dictated by the journal editorial process.

The abstract is too verbose and seems like an introduction.

Response: We reduced the content and changed text to address this comment. Of note, a previous reviewer stated the opposite about the abstract. But we followed your suggestion (lines 17-38).

Mobile apps; blockchain etc., can be examined.

Response: We included an expanded section on mobile apps and the link with telemedicine as it relates to public health information and C19 (lines 599-673 – track changes– all markup or lines 502-572 – track changes – no markup). With respect to blockchain, it is not clear to us how to address this as it is a ledger primarily devoted to commerce and other financial transactions. If you think this is important, please offer some suggestions on how to include it.

Table 1 - too verbose. Organize by keywords or standardized phrases. Also, consider organizing by region, developed v. developing countries, etc., rather a random collection of countries.

Response: The table is organized alphabetically by country, however with the track-changes version it is perhaps not easily noted and confusing to review. The text in the columns is included to generally describe the outcomes. We could include this as text in the body of the manuscript, but felt a table was easier to follow for region-specific information.

Consider discussion political, WHO, limited resources, haves and have nots etc. that have impact on policy.

Response: We chose not to focus on political discussions and these other issues as while interesting and important topics, they are not the aim of this review.

Round 2

Reviewer 1 Report

The paper has been significantly improved following reviewers' comments and suggestions. 

In my opinion, the work is ready to be published.

This manuscript is a resubmission of an earlier submission. The following is a list of the peer review reports and author responses from that submission.

Round 1

Reviewer 1 Report

The paper reviewed the different modalities in approaches to the C19 pandemic by various countries. However, the authors do not clearly present the paper’s main aim and conclusions. It is a mere presentation of measures taken by different nations to manage C19, together with other information such as challenges in various specialities, reorganisation of healthcare systems, and medical staff protection, without a clear motivation.

Overall, the different sections are not well stitched or are not related to each other, as a consequence, it is hard to understand the whole concept.

The introduction jumps from one concept to another, not showing a harmonised argument, in fact, table 1 is not appropriately introduced in any part of the paper. Instead, the text refers to some sheer numbers (line 93) with no reference and no relation to the information in table 1. On the other hand, the information included in table 1 is much more a discussion about the taken measures than an objective list of those measures, which would be expected in the introduction where this table is included.

Table 2 presents measures taken to prevent infection in dental care environments, but it is not referenced in the text nor explained or what is added to the paper.

In section 4, it is not clear what the authors want to express if they are trying to list some proven measures taken to protect medical staff (Telemedicine) or in which areas to be improved (mental burden). There would be a lack of information in both cases as they have provided very superficial reasoning.

Finally, the conclusions section is more a summary than a compendium of results and an appropriate discussion about what they have been adding to the field.

In addition, there is a lack of some references during the text, for example, in lines 106-108; in line 161; or in line 233.

Reviewer 2 Report

Firstly, thank you for opportunity to review very interested article. I don't feel qualified to judge about the English language and style due to not native language.

  1. The title reflect the main subject about adaptation of medical system during COVID-19, title was clear and easy to understand.
  2. The abstract summarize and reflect the work described in the manuscript.
  3. The key words reflect the focus of the manuscript.
  4. The manuscript adequately describe the background, present status, and significance of the study. The authors explain COVID-19 situation in global and adaptation od medical system in many area around the world. However, in part 2.Challenges of the pandemic for different specialties, the authors review many aspects on medical reorganization in specialists but not include emergency department which had many impact about COVID-19 pandemic. In this case I suggested to add some context about that.
  5. The manuscript interpret the findings adequately and appropriately, highlighting the key points concisely, clearly, and logically.
  6. Tables and figures sufficient, good quality and appropriately illustrative of the paper contents.
  7. The manuscript cite appropriately the latest, important and authoritative references in the introduction and discussion sections. However, Some of references were incorrect style for this journal.

Reviewer 3 Report

The present paper provides a review based on a review of previous studies on how COVID-19 infection and the subsequent pandemic caused and had a negative impact on healthcare institutions and practitioners, with potential solutions to these problems. While providing important suggestions, each of the following points (and others, if any) must be improved in order for this to be a review article.

1. The taxonomy in Figure 1 and lines 63-72 of the main text constitutes a basic framework for structuring issues around COVID-19 infection/pandemic, but is too hypothetical and not sufficiently supported by previous research and findings. It is essential to describe more convincingly that these can become the norm of the present paper.

2. Table 1 is an important knowledge base for summarising the status of responses in each country, but the criteria for selecting countries/regions remains unclear. For instance, Japan and Taiwan provide best practices in the East Asia region, but are not mentioned. The fact that there are only 1-2 reference papers for each country is also insufficient. This table is the foundation of the present review article, more comprehensive reviewing of the previous literature is required. As a minor point, California should be deleted; ruled between Australia and Belarus; should realign alphabetically.

3. Table 2 provides measures take to prevent infection in dental care environments, but there is no mention of the citation in the main text and it is abrupt: It is necessary to discuss in the main text why the focus was only on the dental area.

4. Figures 2 and 4 are plain  illustrations thus are inappropriate as figures in a scientific article - should be removed; the same applies to figure 5, which needs to be carefully structured in a more empirical way (with reference to and organisation of previous studies), as telemedicine is an important appeal point of this paper.